# The interplay between vapour, liquid, and solid phases in laser powder bed fusion

I. Bitharas [1 ✉], N. Parab[2,3], C. Zhao [2,4], T. Sun [2,5], A. D. Rollett [6] & A. J. Moore [1 ✉]

The capability of producing complex, high performance metal parts on demand has established laser powder bed fusion (LPBF) as a promising additive manufacturing technology, yet deeper understanding of the laser-material interaction is crucial to exploit the potential of the process. By simultaneous in-situ synchrotron x-ray and schlieren imaging, we probe directly the interconnected fluid dynamics of the vapour jet formed by the laser and the depression it produces in the melt pool. The combined imaging shows the formation of a stable plume over stable surface depressions, which becomes chaotic following transition to a full keyhole. We quantify process instability across several parameter sets by analysing keyhole and plume morphologies, and identify a previously unreported threshold of the energy input required for stable line scans. The effect of the powder layer and its impact on process stability is explored. These high-speed visualisations of the fluid mechanics governing LPBF enable us to identify unfavourable process dynamics associated with unwanted porosity, aiding the design of process windows at higher power and speed, and providing the potential for in-process monitoring of process stability.

[1] Institute of Photonics and Quantum Sciences, Heriot-Watt University, Edinburgh EH14 4AS, UK. [2] X-ray Science Division, Argonne National Laboratory, 9700S Cass Ave, Lemont, IL, USA. [3] Intel Corporation, Hillsboro, OR 97124, USA. [4] Department of Mechanical Engineering, Tsinghua University, 100084 Beijing, China. [5] Department of Materials Science and Engineering, University of Virginia, Charlottesville, VA, USA. [6] Department of Materials Science and Engineering, Carnegie Mellon University, 5000Forbes Ave, Pittsburgh, PA, USA. ✉email: i.bitharas@hw.ac.uk; a.moore@hw.ac.uk

Laser powder bed fusion (LPBF) is the most adopted method for the additive manufacturing (AM) of metallic components, enabling the production of bespoke parts with unparalleled freedom of design and material properties comparable to, or better than, those made using traditional methods[1,2]. However, owing to the high energy typically used in processing, the laser-material interaction results in rapid melting and evaporation of the metal substrate and powder[3]. This interaction is a complex, dynamic process where control over a large number of variables is required for process stability and ultimately, the continuous production of successful parts over time or across build platforms.

Fundamental understanding of the physics involved in additive manufacturing can aid in tuning these processing parameters. It has been shown that the behaviour of the melt pool and induced vapour jet is highly variable, leading to several different hydrodynamic regimes[4–6]. Moreover, those regimes also determine the interaction with the powder particles, which is a critical component of process stability[5,7–9], influencing part quality and structural defects (e.g. porosity)[10]. While great progress has been made to understand and characterise these dynamics by imaging the behaviour of the liquid metal[6,11–13], powder particles[8,9,14] or vapour[3,15] individually, their interplay and combined motion has not been observed directly.

In this paper, we present simultaneous schlieren and x-ray transmission imaging, which has enabled the visualisation of the interaction between all phases of matter in LPBF at the same time. In particular, the high magnification of the schlieren system allowed direct imaging of the vapour jet emerging from the melt pool depression at varying laser parameters; we reveal that the onset of instability within the keyhole causes a transition from stable to chaotic flow in the laser plume. This transition causes refractive index changes in the atmosphere, that are measurable even if x-ray imaging is not available. Systematic image analysis allowed us to identify a threshold input energy density for the onset of this instability in Ti–6Al–4V alloy. We found that introducing the powder increases the keyhole stability under the examined laser conditions and slightly raises the threshold input energy density. This imaging study allows more intuitive

understanding of the fluid dynamics governing LPBF, aiding in the interpretation of in situ diagnostics, neural networks, and numerical simulations of LPBF, by establishing the interconnectivity between the melt pool and the vapour jet, and their combined effect on the plume.

## Results

**Coupling of melt pool and plume dynamics.** To image the motion of liquid metal and the depression caused by the released vapour, high-energy synchrotron x-rays passed through ~400 μm thick Ti–6Al–4V samples, held in place by glassy carbon slides inside the LPBF process simulator at the 32-ID-B beamline of the Advanced Photon Source[16]. The motion of the vapour and airborne particles in the Ar atmosphere were visualised by schlieren imaging[15], via pick-off mirrors (M1, M2) that folded the optical beam at an angle $\varphi \sim 2°$ to the x-ray beam, resulting in a nearly co-axial (to the x-ray beam) view of the process (Fig. 1).

Experiments were first conducted on the Ti–6Al–4V substrate only to provide a baseline comparison of the process without powder. Figure 2 shows the evolution of a surface depression and associated vapour jet and plume under stationary laser illumination ($\lambda = 1070 \pm 10$ nm) for power density $\Phi = 1.3$ MW cm$^{-2}$. Initially, the surface under the laser spot rapidly heats up over the boiling point of Ti–6Al–4V (3133 K)[17,18], the vapour pressure of Al and Ti increases, and a vapour jet is released upwards. Figure 2a shows the propagation of this flow in the Ar atmosphere as it entrains the surrounding gas, forming a characteristic laser plume. Refractive index gradients outline the interface between the plume and the atmosphere, which are proportional to density gradients in the fluid. These density gradients are caused by the underlying pressure and temperature fields, in addition to the concentration of the evaporated species[19]. The schlieren features appear darker or lighter than the background, indicating a change in sign of the gradient. At the bottom of the image, between the positive and negative refractive index gradients, the vapour jet can be discerned at the core of the plume. As the hot metallic vapour ejected from the liquid-vapour interface (LVI) cools, nanoparticles form[20,21]

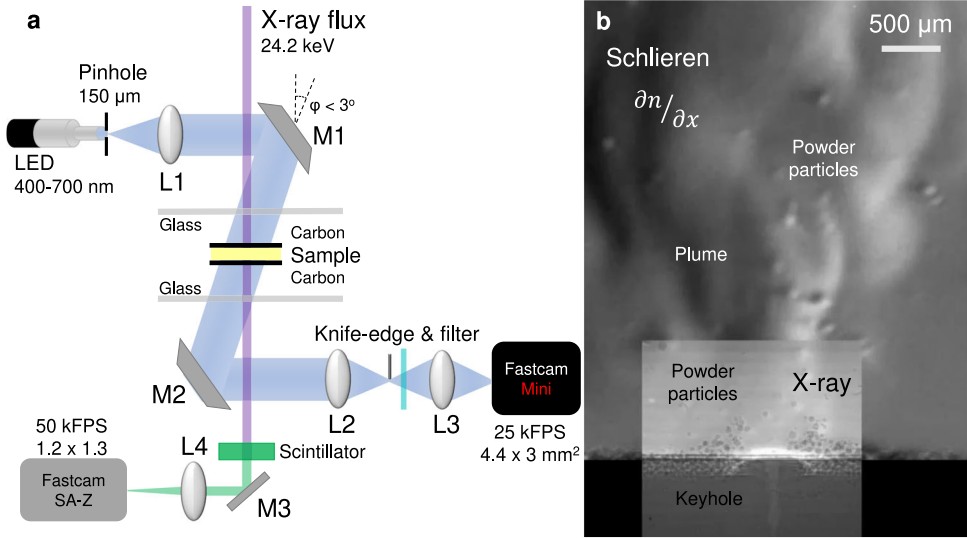

**Fig. 1 Overview of simultaneous optical and x-ray experimental setup. a** Schlieren setup: L1: SMC Pentax-A 50 mm F1.7, L2: Sigma DL 75-300 mm, L3: Sigma 150-600 mm f/5-6.3. The knife edge was set to block 50 % of the incoming light, evenly measuring refractive index gradients $\partial n/\partial x$. Low-pass filtering (Schott KG-5 glass) blocked excess process light. X-ray transmission setup: A Lu$_3$Al$_5$O$_{12}$ scintillator converted the x-ray signal to 535 nm light, M3: flat mirror, L4: 10x Objective (N.A. 0.28). **b** Composite image comprising superimposed x-ray and schlieren images of Ti–6Al–4V sample and ~100 μm height powder layer processed in an Ar atmosphere within the LPBF simulator.

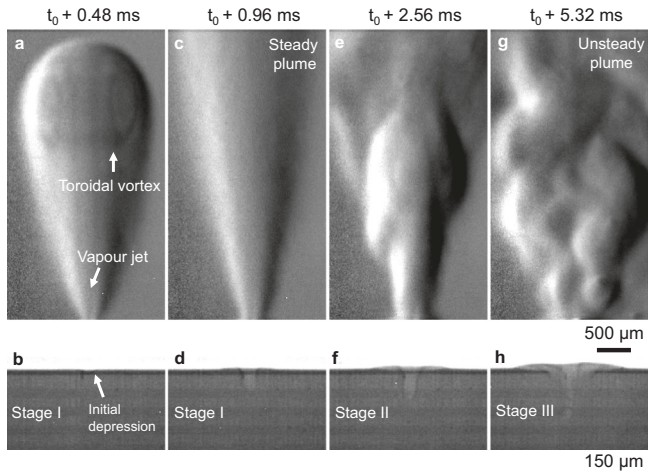

**Fig. 2 Time evolution of depression and plume under a stationary laser spot of power $P = 72$ W and $1/e^2$ diameter $d = 84$ μm, corresponding to a power density of $\Phi = 4\,P/\pi d^2 = 1.3$ MW cm$^{-2}$ for substrate only (no powder).** Time evolution of depression and plume under a stationary laser spot of power $P = 72$ W and $1/e^2$ diameter $d = 84$ μm, corresponding to a power density of $\Phi = 4\,P/\pi d^2 = 1.3$ MW cm$^{-2}$ for substrate only (no powder). **a** Initial plume front propagating upwards in Ar atmosphere due to vapour jet. Metallic fumes follow the flow's streamlines, outlining a vortex structure at the tip of the plume. **b** Initial depression (depth ~ 5 μm). **c** Expansion of the evaporated species leads to a steady atmospheric flow. **d** Depression depth steadily increases (depth ~ 25 μm). **e** The momentum driving the flow changes direction according to the motion of the vapour jet, intermittently disrupting flow stability. **f** The depression expands rapidly, while the previous hemispherical shape is maintained (depth ~ 120 μm). **g** Chaotic plume comprised of several eddies formed by constant fluctuation of the LVI. **h** Fully evolved keyhole, driven by the interplay between recoil pressure and capillary instability. 500 μm scale bar applies to (**a**, **c**, **e**, **g**), 150 μm scale bar applies to (**b**, **d**, **f**, **h**). A video with the images used in this figure is available in the supplementary materials.

via nucleation and condensation, growing in size through processes such as continuous oxidation, coagulation and agglomeration[22–24] as they propagate upwards. A high concentration of nanoparticles (i.e., fumes) blocks the incident broadband light of the LED, producing the dark lines that trace the fluid motion. At the top interface between the ambient medium and the plume, a toroidal vortex can be observed due to the suspended particles, validating recent numerical simulations[25]. This structure, similar to a mushroom cloud from an explosion or a gas bubble rising within a liquid, is mainly the result of viscous stresses developing between the fast-moving fluid and the static atmosphere, causing the flow to curl outwards along the direction of motion[26,27]. Figure 2b shows the corresponding x-ray image for that time instant; only a slight surface depression has been formed on the molten volume due to the recoil of the vapour pressure. It is interesting to note that a substantial plume is present prior to any significant surface depression, owing to the high speed of the vapour jet[3,15,18,28,29] of the order of hundreds of m s$^{-1}$. The vapour jet dissipates, entraining the surrounding Ar atmosphere to create a plume that rises due to the net accumulated momentum[15,25].

After a further 0.48 ms of laser irradiation, the surface depression continues to deepen (Fig. 2d). Its depth progression is gradual with a parabolic shape because of the vapour's pressure distribution across the surface. The jet of evaporated species emitted from that surface is in turn stable, and therefore the atmospheric flow appears constant (Fig. 2c); this stable plume

expands as it propagates upwards. Over time, the surface depression deepens (Fig. 2f), and oscillations in the LVI mark the transition to a keyhole. The vapour emitting surface of the keyhole is perturbed, changing the direction of the vapour jet and plume (Fig. 2e). Sudden changes in the keyhole shape correspond to more pronounced fluctuations of the flow structure. The intensity of the density gradients of Fig. 2e is noticeably higher compared to previous frames: the background intensity remains unchanged compared to previous frames, indicating that the temperature and concentration of the vapours has increased. This observation is consistent with measurements of increased laser absorption as the depression depth increases[30–33] and calculations of high temperature regions on the keyhole walls as the laser drills into the material[29].

Figure 2h shows a fully formed keyhole, the surface of which is constantly fluctuating according to the interactions of vapour pressure, surface tension, gravity, drag from the vapour flow, and accumulated momentum within the liquid metal volume[30,34,35]. As a result, the plume is disrupted completely (Fig. 2g), and a more chaotic flow is observable. Despite this instability, momentum is still predominately directed upwards, but the flow (and fume distribution) is more diffuse due to the formation of eddies.

While the formation of eddies and chaotic mixing observed in the atmosphere partly satisfy the criteria of turbulence, the spatial resolution of the schlieren system was insufficient to characterise the finer flow structures, such as those at the core of the vapour jet or the stream of fumes and therefore insufficient evidence to characterise the flow at any time as "laminar" or "turbulent". Furthermore, it is difficult to predict the behaviour of the evaporated jet in terms of Reynolds number ($Re$), as lack of reliable thermophysical properties for Ti and Al vapours prevent any meaningful $Re$ estimates for Ti–6Al–4V, in addition to variability in the dynamics of microjets[36,37] further complicating its interpretation. Nevertheless, numerical modelling of Fe vapours in LPBF[15] has produced estimates of $Re \sim 300$ because of the high velocity and temperature, but small jet diameter; this is in agreement with experiments[38] showing that subsonic microjets experience a sudden breakdown after a short propagation length at $Re > 450$, which was not observed in our experiments.

Figure 2 indicates that the dynamic melt pool behaviour can be divided into three stages, corresponding to the stable formation and growth of the surface depression and laser plume (stage I), which transitions into a metastable state where the LVI is perturbed but its shape recovers (stage II) and ultimately, unstable melting and plume due to a continuously fluctuating keyhole (stage III). We purposely avoid using the terms conduction, transition and keyhole regimes[1,6,11,30], which are often used to characterise ex situ micrographs of the melt pool in both laser welding and LPBF[39,40] but which can be misleading for these in situ keyhole and laser plume measurements. A conduction mode melt pool contains a stable 'keyhole' depression[4,6] even for power densities much lower than those typically used in LPBF. As the power density increases, the stable depression increases in depth to produce a transition mode melt pool that is elongated in the depth direction[39], and eventually an even deeper keyhole mode melt pool[6,11], although there is no single measure where that begins. There is likely to be some overlap in power density range between stage II describing the onset of keyhole and laser plume instability and the regime for a transition mode melt pool cross-section, but there is no expectation or requirement that these qualitative descriptions of different aspects of the process correspond exactly.

The evolution of the keyhole was examined by measuring the depression depth in the x-ray images, as well as the average

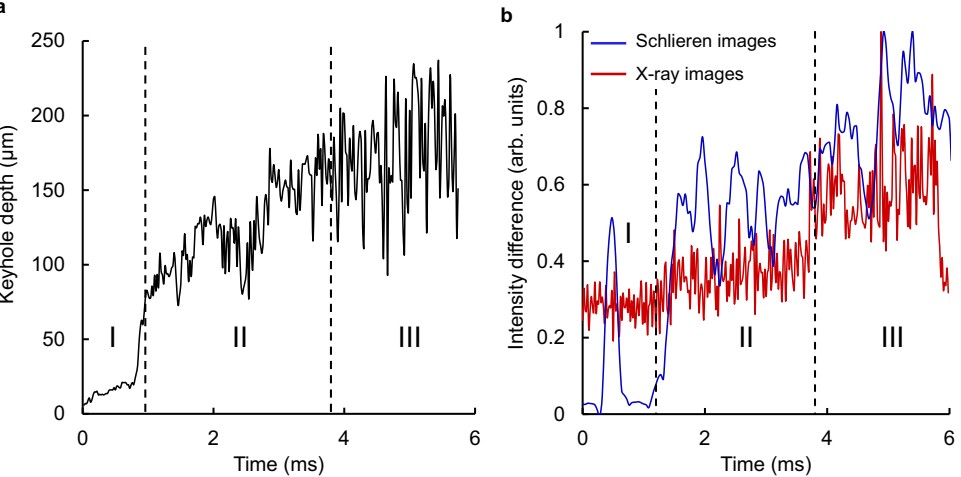

**Fig. 3 Analysis of vapour depression evolution. a** Measured surface depression depth. The keyhole depth increases gradually during stage I, with a rapid drilling marking the transition to stage II. The penetration fluctuates increasingly over time, due to keyhole instabilities. **b** For both datasets, the total change in each frame increases over time as the system tends towards instability. Stages I–III of keyhole evolution are discernible in both datasets, owing to the strong coupling of liquid-vapour phenomena.

intensity difference (as outlined in the Methods section) between consecutive images in both the x-ray and schlieren datasets (Fig. 3). The continuous increase in variability, both in depth and in the intensity differences, is indicative of the melt pool's progression towards instability over time. The stages of the melt pool can be distinguished, as instability levels in the plume register distinctly (Fig. 3b), matched by the increased keyhole oscillations. The strong coupling of the liquid and vapour phases is thus evident in the synchronisation between periods of stability and instability. The isolated peak in stage I for the schlieren data is due to the initial establishing of the laser plume seen in Fig. 2a. Under a stationary laser, the melt pool will always undergo these stages, provided that the laser power density is sufficiently high; naturally, the time required for the keyhole to form and eventually become unstable is dependent on the total energy input[6,11].

**Input energy density threshold for keyhole stability**. Scanning the beam introduces additional dissipation of the laser energy and changes the momentum balance in the melt pool, which affects the stability of the melt pool[12]. Simultaneous imaging was carried out for line scans on the substrate without powder for varying power density and scan speed (Fig. 4). For each laser power, the figure shows two consecutive frames captured at time $t_1$ and $t_1 + 40$ μs, where the melt pool has reached a steady state after a scan length of ~2 mm, in order to demonstrate the relative stability of the keyhole depression and laser plume in each case. The figure also indicates the linear input energy density for each laser line scan, defined as $E = 4P/\mathbf{v}\pi d^2 = \Phi/\mathbf{v}$, where $\mathbf{v}$ is the laser scan speed and $\Phi$ is the previously defined power density. We show here that this energy density can be used as a meaningful thermodynamic metric for LPBF, allowing to estimate a threshold beyond which the vapour depression becomes unstable. It is worth emphasising that, despite the common units of J m$^{-3}$, this is not a volumetric energy density typically defined using the hatch spacing or powder layer thickness[40].

When the laser power is relatively low and the scan speed relatively high (Fig. 4a), the energy density $E \sim 27$ GJ m$^{-3}$ is dissipated fast enough that the melt pool remains at stage I; the surface depression is shallow, while its shape remains constant. The vapour jet is visibly established perpendicular to the irradiated surface, which is mainly the front wall of the

depression. Owing to the invariance of the LVI, the jet's angle and induced atmospheric flow remain constant over time. As the input energy density increases to $E = \sim 69$ GJ m$^{-3}$, the keyhole is deeper and longer but retains its shape, with periodic fluctuations of the rear wall (Fig. 4b). The depression is only perturbed weakly, as the heat dissipation by the scanning motion prevents the LVI from becoming unstable; thus, the plume's steady flow structure is preserved. Due to the higher laser power, a higher vapour content is observable within the plume and a bright region appears over the keyhole. This bright region is attributed to thermal radiation emitted by hot vapour and condensate (fume)[15]: reflections or scattering effects are ruled out because 1070 nm radiation was heavily filtered by the KG-5 glass.

In Fig. 4c, the laser power density is reduced by 33% but the scan speed is reduced by 45 % compared to Fig. 4b, resulting in a higher energy density $E = \sim 82$ GJ m$^{-3}$. A stage III melt pool is observed in this case, with stronger oscillations, frequent collapses of the keyhole walls and several disjointed density gradients in the laser plume due to the complex vapour flow pattern emerging from the keyhole. Increasing the incident laser power density by a factor of ~2.1 resulted in an even more dynamic melt pool with a deeper, intensely oscillating keyhole (Fig. 4d). The white streak observable in the laser plume at $t = t_1$ is again attributed to thermal radiation due to interaction of the incident laser beam with the fume.

Many experiments similar to Fig. 4 were conducted with combinations of laser powers and scan speeds in the range 203–442 W and 0.2–1.5 m s$^{-1}$, respectively, for $1/e^2$ beam diameters of 84 and 99 μm. Automated edge detection of the keyhole boundaries enabled its area to be calculated in every image. The average area, with error bars representing its standard deviation through all images in each sequence, is plotted against the energy density in Fig. 5a. The area measurements are indicative of the wide range of possible keyhole morphologies, with relatively large depressions forming even at low input energy densities. However, the rapid increase in standard deviation relates directly to the increased instability of the melt pool and of the LVI. The degree of instability was again quantified from the intensity difference between consecutive images of the keyhole and laser plume, averaged through all the difference images in each sequence (Fig. 5b). For the x-ray data, the average intensity difference increases linearly, as indicated by the least squares fit, showing that instability levels scale with input energy density.

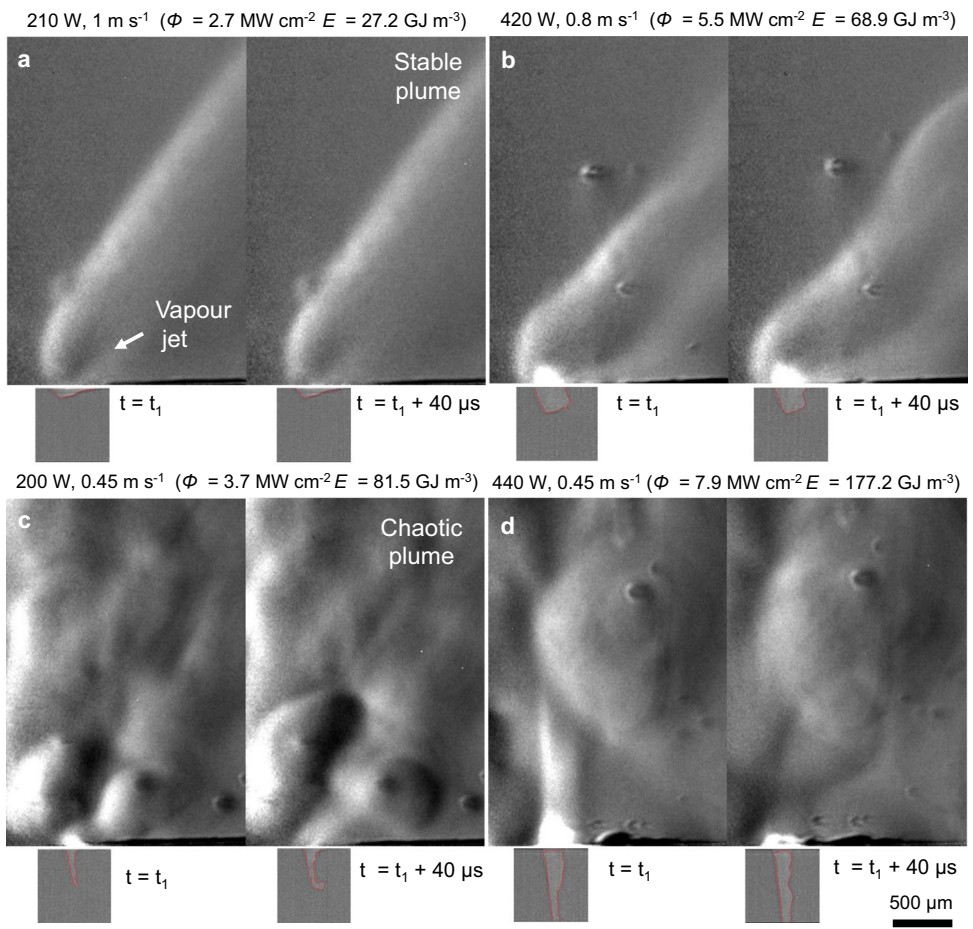

**Fig. 4 Visualisation of the Ti–6Al–4V substrate only (no powder) and Ar atmosphere during scanning of single lines with varying input energy densities.** The outline of the keyhole boundary is highlighted in red in the x-ray images, which are at the same scale as the schlieren image. **a** Under low energy input, the depression's shape remains constant and a steady vapour jet emitted from the irradiated surface results in a steady plume. **b** For $E < 70$ GJ m$^{-3}$ the depression remains at stage II, where small fluctuations in the LVI perturb the plume but the flow pattern remains. **c–d** High input energy density due to (**c**) low scanning speed or (**d**) high power density result in stage III depressions with chaotic plumes. A video with the images used in this figure is available in the supplementary materials.

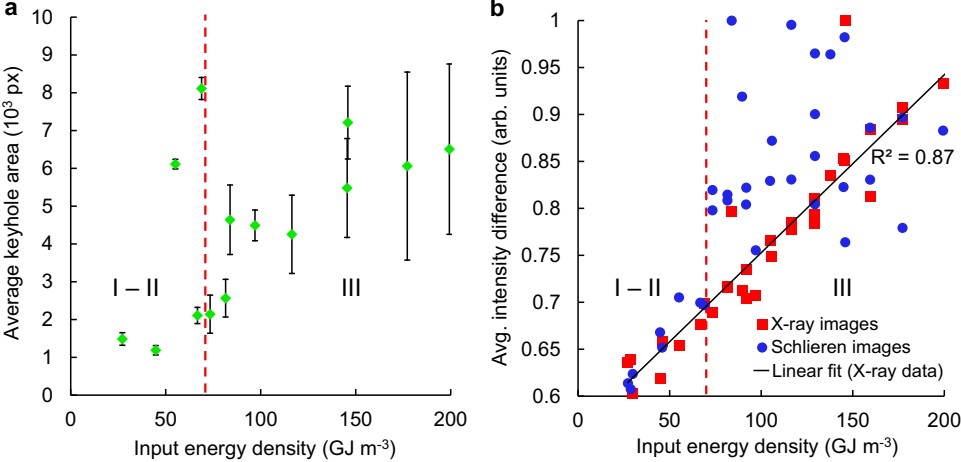

**Fig. 5 Measurements of melt pool/plume stability under varying input energy density. a** Plot of keyhole area (error bars: standard deviation $\pm\sigma$), showing the melt pool becomes increasingly unsteady as the input energy density increases. **b** Average intensity difference between consecutive frames. The red dotted line indicates a step increase in average intensity difference for the schlieren images at $E = {\sim}70$ GJ m$^{-3}$ that marks the transition from stable to chaotic plumes, and thus, depression instability.

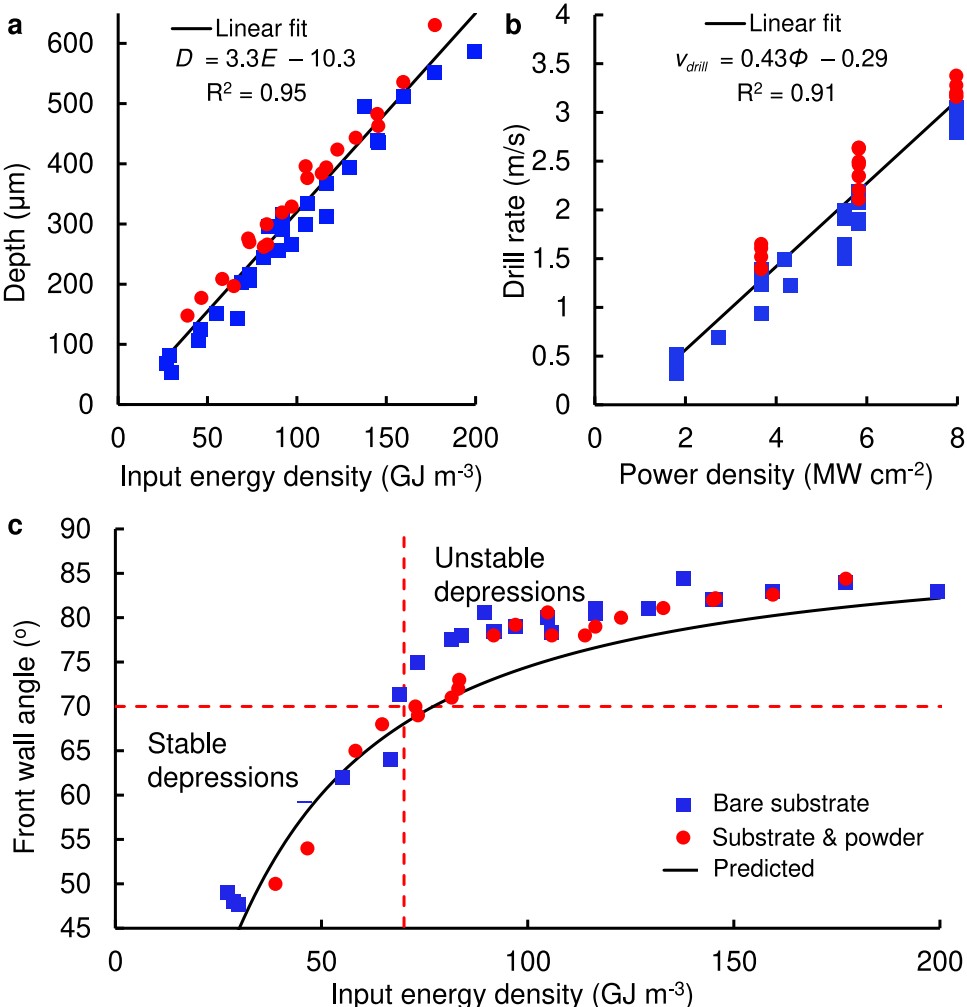

**Fig. 6 Measurements of depression morphology during laser line scanning for experiments without and with powder. a** Depression depth variation with input energy density. **b** Calculated drill rate based on experimental depth. **c** Measured and predicted front wall angle, according to $\tan\theta = D/d$. All three graphs use the same legend shown in (**c**).

However, for the schlieren data, the variability only increases linearly for experiments at $E < 70\,\mathrm{GJ\,m^{-3}}$ beyond which it undergoes a considerable increase and is no longer linearly related to the energy density. Experiments with $E < 70\,\mathrm{GJ\,m^{-3}}$ had steady plumes and depressions, while chaotic flows with unsteady depressions were observed when that energy was exceeded.

The definition of input energy density we use enables a greatly simplified approach to predict the depression size and provides a physical basis for the threshold in keyhole stability observed at $70\,\mathrm{GJ\,m^{-3}}$. It has been shown previously that the depression penetration depth, $D$, scales linearly with laser power, but a different line was obtained at each combination of scan speed and beam diameter[6,13]. A geometrical model relating this penetration to the angle of the front wall of the keyhole, $\theta$, has also been validated[4,6]. In that model, $\tan\theta = D/d = v_{\mathrm{drill}}/v$, where $v_{\mathrm{drill}}$ is rate at which a stationary laser spot drills into the material. Plotting the front wall angle against power density again produced a separate line for each combination of scan speed and beam diameter[6]. Hence it is difficult to use this information for parameter selection, or to explain melt pool regime change across the parameter space.

The keyhole depth and front wall angle were measured in the x-ray images and are plotted against input energy density (Fig. 6a, c). The measured depth was used to calculate the drill

rate for the corresponding power density range (Fig. 6b). The straight lines represent a least squares linear regression to the depth and drill rate. Using the gradient of the fitted lines, the front wall angle at each energy density can be predicted: a least squares fit to those points is the line shown in Fig. 6c. Immediately it is seen that the input energy density provides a single line for the depth and front wall angle across all the laser powers, scan speeds and beam diameters tested, which can now be used for process parameter selection.

Time-averaged measurements[32] of the absorptivity $A$ of Ti–6Al–4V have shown that for $15 \leq E \leq 200\,\mathrm{GJ\,m^{-3}}$, $A$ increased from 0.3 to 0.7, while the total absorbed energy density $EA$ increased linearly with $E$ (see supplementary materials). The observed linear scaling of the depression depth with $E$ (Fig. 6a) suggests that the additional absorbed energy compensates for larger energy losses by the various thermophysical and hydrodynamic effects in the melt pool as $E$ increases. We observe a steep increase in the depression's front wall angle close to $E = \sim70\,\mathrm{GJ\,m^{-3}}$, where the front wall angle approaches 70°. In light of the stability analysis (Fig. 5), this rapid rise in front wall angle is characteristic of the transition to a stage III depression. This measurement is consistent with recent numerical modelling of keyhole dynamics[29], showing that an increase in inclination from 65° to 75° initiates the occurrence of multiple reflections and dynamic change of local laser absorption within a scanning depression, causing the complex

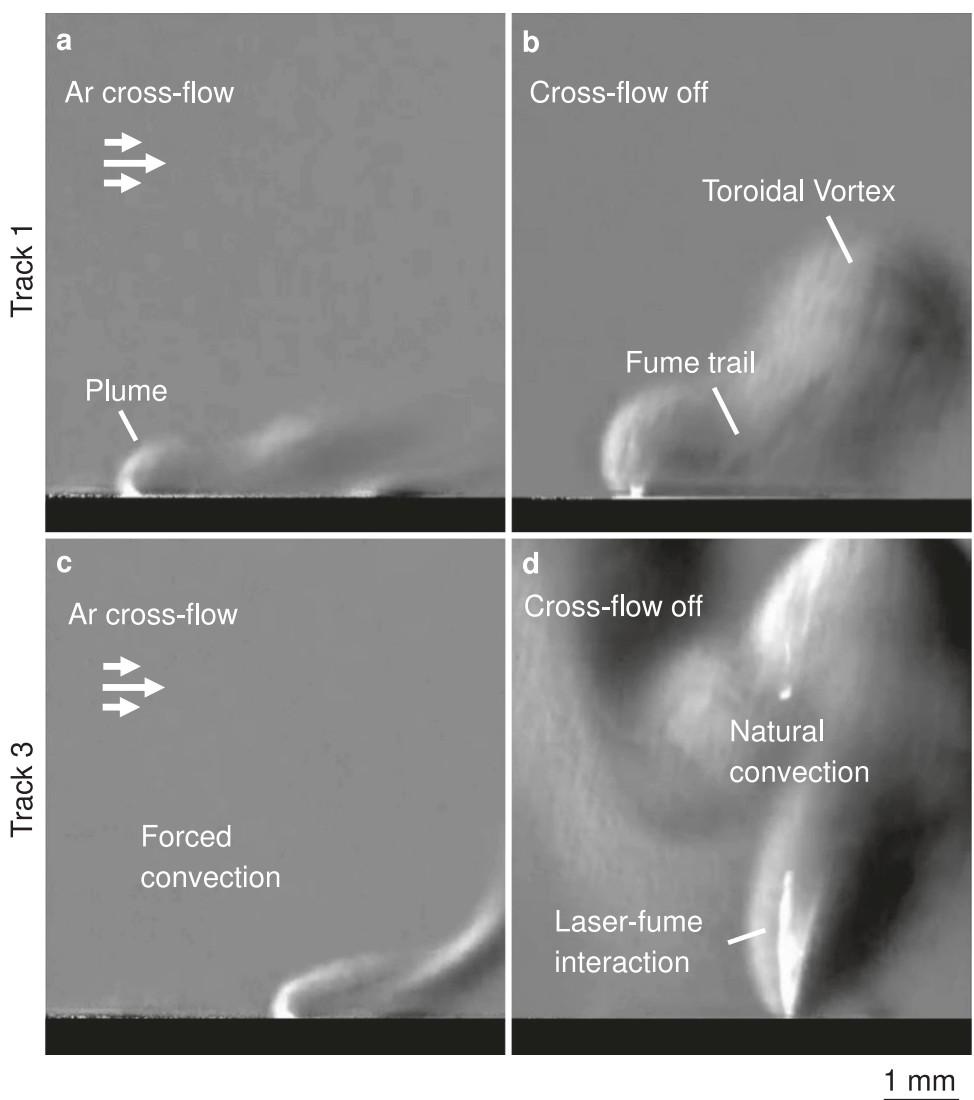

**Fig. 7 Schlieren images of the Ar atmosphere above SS316 samples during 5 × 5 mm² island scans, with and without cross-flow (peak *Re* ~ 1400, *u*ₘₐₓ ~ 2.1 m s⁻¹).** In all images, the laser is moving right to left, **v** = 0.75 m s⁻¹. Refractive index gradients outline the plume above the melt pool regardless of the cross-flow, due to the strong evaporation from the depression. **a**, **b** Track 1 of the scan: the laser-material interaction is similar in both cases, as the laser processes through a clear atmosphere. **c**, **d** Track 3 of the scan: c By-products from earlier tracks are extracted with the cross-flow on, promoting repeatability. **d** Frequent laser-fume interactions were observed when processing in a fume-rich atmosphere.

thermocapillary phenomena and vapour recoil pressure responsible for unstable keyhole formation.

**Influence of Ar cross-flow.** During LPBF, interaction of the laser beam with process by-products such as fumes and spattered particles can result in defects in manufactured parts[15,41,42]. For this study, line and spot welds were carried out in a static Ar atmosphere, using the schlieren system to ensure no fumes were present prior to each experiment. However, in commercial LPBF machines, a laminar Ar cross-flow is often introduced to extract process by-products away from the scanning area. To avoid disruptions to the powder layer and atmospheric turbulence, typical peak velocities for this flow are of the order of a few m s⁻¹, and therefore the convective cooling and stagnation pressure applied to the melt pool are second-order effects in comparison to the forces arising from the laser-material interaction[15,18]. Consequently, this flow has been largely omitted from numerical meso-scale models, while experiments not carried out in

commercial LPBF chambers often rely on natural convection for fume extraction. However, as the dynamics in this study pertain to atmospheric effects above the melt pool, the effects of this flow must be considered to establish the applicability of the results to the actual LPBF process.

Figure 7 presents exemplary data on SS316 recorded in an open architecture LPBF system[15] in a separate experiment. The same schlieren apparatus reported here was used but set to a wider field of view so that the effect of the cross-flow on the laser plume is visible. An Ar cross-flow with a peak velocity $\mathbf{u}_{max}$ ~2.1 m s⁻¹, and a peak Reynolds number of 1400 (defined as $Re = \rho \mathbf{u}_{max} D / \mu$, with the hydraulic diameter $D = 0.01$ m equal to the cross-flow device's outlet area divided by its wetted perimeter, and $\rho = 1.62$ kg m⁻³, $\mu = 23.6$ μPa s) was used. With the cross-flow (Fig. 7a), the plume is dissipated within the convective flow after propagating upwards for ~1 mm, where its local momentum approaches that of the Ar stream. Without a cross-flow (Fig. 7b), the plume rises higher, until its velocity roughly matches the scan speed of the laser. A trail of fumes is left at its wake, rising under

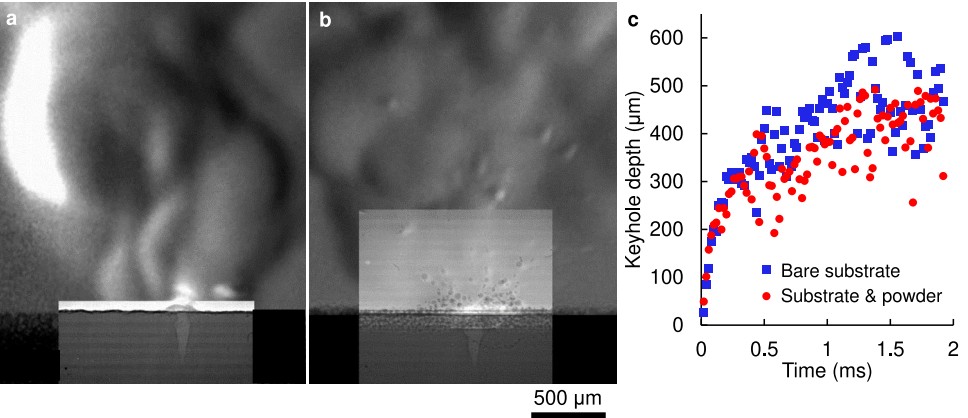

**Fig. 8 Comparison of laser-material interaction at high power.** Composite images of keyhole and plume (**a**) without and (**b**) with powder under 3.7 MW cm$^{-2}$ laser power density ($P = 204$ W and $d = 84$ μm). Both cases were qualitatively similar, with a high drill rate resulting in the formation of an unstable keyhole. The penetration depth (**c**) was consistently lower for the powder case, due to a more dynamic upper keyhole as well as laser-particle interactions. A video with the images used in this figure is available in the supplementary materials.

buoyancy and its remaining momentum. During this first line scan with no cross-flow, the atmosphere ahead of the laser is clear, and therefore the interaction between the laser and the solidifying vapour is minimal. The similarity of imaged features with and without the cross-flow (Fig. 7a, b) demonstrates that the cross-flow does not significantly change the laser-material interaction in a clean atmosphere, and so the results presented herein are applicable in both cases. The situation is clearly different for subsequent tracks in an island scan. With a cross-flow, products from preceding tracks are convected away, resulting in identical process conditions to the first track throughout the scan (Fig. 7c). Without a cross-flow, laser-fume interactions were observed frequently as the island scan proceeded due to the accumulation of vapour and fume from the preceding tracks over the sample (Fig. 7d). Note that the simultaneous x-ray and schlieren imaging were not conducted in these conditions.

The similarity between Fig. 7a, b can be attributed to the vapour jet's high momentum. Its peak velocity is of the order of hundreds of m/s[15,18,25,28] while that of the Ar stream peaks at ~2.1 m/s and is reduced closer to the sample due to the laminar flow's profile and boundary layer effects. Therefore, minimal skewness or disruption in the laser plume close to the melt pool is predicted by numerical modelling[15]. While it is possible that the cross-flow could partly obscure some of the dynamics imaged in this study, refractive index gradients from the vapour jet and plume are observable under all conditions due to its high temperature and metallic species content close to the substrate. Therefore, although the plume stability threshold in the present study is identified in the preceding section in a stationary atmosphere, it is expected that the plume could provide enough information to characterise the melt pool stability in situ with a cross-flow, provided the spatial and temporal resolution of the measurement were comparable to those used here. Alternatively, a static atmosphere could be used as part of a calibration process, where the effects of single scans with varying input energy density can be interrogated fully, before extrapolating to part-scale processing with the cross-flow enabled.

**The effect of powder particles.** To visualise the interaction between the LVI and solid particles, spot welds and line scans were carried out using a ~100 μm powder layer (spherical Ti–6Al–4V particles of 15–45 μm diameter, spread manually with no compaction on pre-aligned Ti–6Al–4V coupons) under varying energy input. In all experiments, the top surface of the

powder layer was positioned at the same height as the top surface of the samples without powder, in order to maintain a common reference plane for the calibrated laser spot diameter and the measured keyhole depth between the two cases.

Under a stationary laser power density of 3.7 MW cm$^{-2}$, the depression initially has the same drill rate regardless of the powder layer, transitioning to an unstable keyhole after ~350 μs (Fig. 8). During the initial period, the intense drilling interaction rapidly melts, evaporates, or ejects spatter and particles alike. In both cases, the penetration depth fluctuated significantly over time, due to the hydrodynamic instabilities of the keyhole regime, resulting in the immediate formation of a chaotic plume. Powder particles were continuously entrained from the vicinity of the laser spot due to the induced atmospheric flow[3,14,43]. A fraction of these particles was incorporated into the melt pool, while the rest were lifted off the powder bed. It is clear from the composite images that particles near the powder bed that entered the visualised plume area were ejected by lift forces from the jet[15]. Lower penetration was observed when powder was present (Fig. 8c). We attribute this reduction in penetration to a combination of two factors in the powder case: i) the porosity of the powder layer and particles ejected by interacting with the vapour jet/plume result in an inconsistently shaped upper keyhole, allowing a larger fraction of the incident light to escape the cavity with less interaction, thus lowering the effective absorptivity, and ii) ejected powder particles periodically block incident light, slowing the accumulation of energy in the melt pool.

Figures 2 and 3 showed that the laser power density of 1.3 MW cm$^{-2}$ resulted in a more a gradual stage I – III transition than for 3.7 MW cm$^{-2}$ (Fig. 8). The laser-material interaction at this lower power density differed significantly from the substrate only (Fig. 9a) with the introduction of powder (Fig. 9b, c): powder particles were continuously consolidated in the melt pool without forming a keyhole, while a slower vapour jet ejected fewer particles, resulting in a large globular melt pool. The recoil pressure of the vapour jet caused a sideways motion of this molten volume. The composite image shows the plume establishes perpendicular to the surface at the exposed spot, indicating the direction in which the reaction force is applied onto the melt pool[3,6,44]. The molten mass and plume oscillated continuously while exposed to the laser; this motion enhanced powder denudation, resulting in the incorporation of more powder particles and increase in melt pool size over time.

The apparent instability can be explained by considering the forces acting on the surface of this globular melt pool. Where the laser is incident, the recoil pressure pushing the surface is in

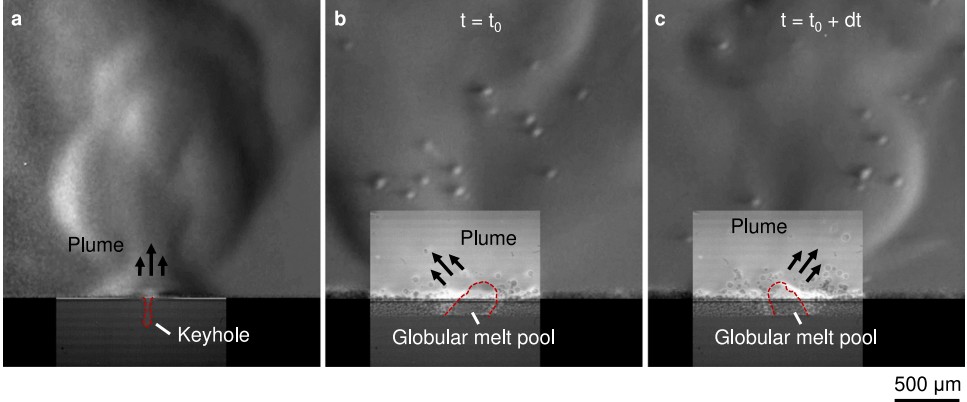

**Fig. 9 Comparison of laser-material interaction at low power.** Composite images of keyhole and plume (**a**) without and (**b**, **c**) with powder under 1.3 MW cm$^{-2}$ laser power density ($P = 72$ W and $d = 84$ μm). The introduction of powder prevents the evolution of the depression into a keyhole. Instead, a large melt pool is formed, with the vapour jet moving the surface depression, thus displacing the molten volume. The direction of the plume continuously shifts from left to right as the interplay of surface forces drive a periodic motion of the liquid metal. The refractive index gradients in the observed plume are weaker in all cases compared to Fig. 8, owing to the lower concentration of evaporated material at this lower powder density. A video with the images used in this figure is available in the supplementary materials.

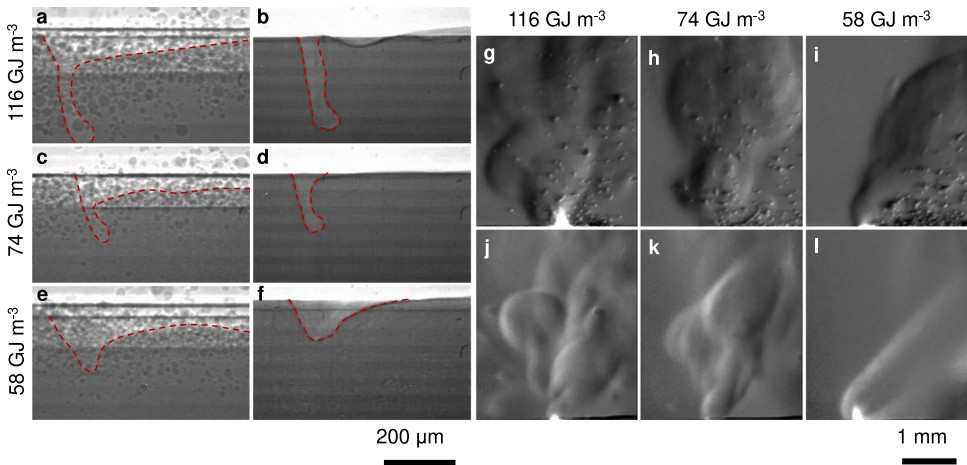

**Fig. 10 Comparison of images of line scans with powder and without powder at varying input energy density. a–f** X-ray images, showing that regardless of energy, melting powder on top of the substrate does not produce large differences in the surface depression, because the laser beam interacts mainly with molten material. **g–l** Schlieren images reveal how the plume and atmospheric flow interacts with the powder layer. 200 μm scale bar applies to panels (**a–f**), 1 mm scale bar applies to panels (**g–l**). A video with the images used in this figure is available in the supplementary materials.

competition with the local Marangoni force and capillary pressure[28,30,34,35]. When the vapour pressure is low, the downwards drill rate is comparable to the sideways motion of the liquid, resulting in a diagonal motion of the depression. The observed motion shows that upper part of the LVI of a large semi-spherical molten mass is unstable, resulting in the movement of the liquid volume. We hypothesise that, even if the liquid droplet is stationary and symmetrically disposed underneath the laser beam, any perturbation to one side tilts the recoil pressure direction so as to move the droplet further off centre. In so doing, the liquid accelerates until the restoring force of the excess LVI being generated brings it to a halt. The momentum of the liquid has, however, carried it beyond the neutral point leaving a net restoring force from capillary action. The droplet accelerates back and once again momentum carries it past the symmetric point thereby allowing the cycle of oscillation to repeat. This oscillating instability was consistent across all experiments carried out with lower power, where the penetration depth in the bare plate did not rapidly exceed the depth of the powder layer. Whilst the motion of this globular melt pool under spot illumination is not directly applicable to the majority of LPBF processes, the observed dynamics demonstrate that the total powder mass incorporated into a melt pool influences the depression's morphology, which in turn affects the stability of all LPBF processes.

Figure 10 shows frames from line scans with input energy densities in the range 58–116 GJ m$^{-3}$. The x-ray images show that powder particles are continuously incorporated into the melt pool during translation of the laser beam. The schlieren images reveal that, in addition to guiding entrained particles towards the laser beam[3,15], the atmospheric flow induced by the vapour jet affects the local availability of powder. When the laser scan speed is low, the interaction time between the atmospheric flow and powder particles is longer, while the vapour jet velocity is higher because of the higher keyhole temperature[3]; as a result, a larger fraction of nearby particles are ejected[15], increasing the amount of hot spatter (Fig. 10g). In addition, the chaotic flow and highly variable vapour jet result in a wide range of ejection angles for hot spatter. Conversely, a constant depression with a shorter interaction time results in airborne particles with more homogeneous trajectories (Fig. 10i).

The observed keyhole morphologies were qualitatively similar regardless of the presence of powder. The lower mass in the powder layer results in a consistently deeper penetration, compared to the substrate only, across the full range of line scan input energy densities (Fig. 6a). The slope of the linear depth increase against increased energy density is similar in both cases. The plume follows a similar progression towards instability with increasing input energy density, with and without powder. The refractive index gradients appear stronger and slightly more irregular with powder due to stronger evaporation from heated particles. The ejection angle of the plume for powder shows the same increase with increasing input energy density, as expected from the increasing front wall angle. The measured front wall angle with powder (Fig. 6c) follows a similar trend to that for the substrate only. An angle of 70° was reached at an energy density slightly higher than 70 GJ m$^{-3}$, indicating that the powder layer reduced the overall energy absorption by the sample. As discussed earlier, the upper keyhole is wider than the bare substrate case, which reduces the laser reflections within the cavity, whilst powder particles (that absorb laser energy) are often pushed away instead of being incorporated into the melt pool.

## Discussion

The visualisations presented in this article show the interconnectivity between the evaporation dynamics, motion of liquid metal and particle behaviour in LPBF. We observed that the behaviour of the surface depression is shown in the motion of the vapour jet and plume, which implies that the stability of the system can be determined by interrogating either. In situ x-ray imaging is only possible at the few light sources that can deliver high energy and high intensity x-rays: this work shows that plume monitoring is a far more generally accessible alternative to probe the behaviour of the keyhole and melt pool. Moreover, we revealed that the transition to unstable keyhole is accompanied by a transition to chaotic flow in the plume, which serve as a distinct signature that could possibly be monitored and integrated in closed-loop control systems. Such systems could also make use of subtler features in the plume, such as the observed larger refractive index gradients with an increase in laser drill rate.

The intensity difference between consecutive frames of schlieren images was characteristic of the stability of the process, which enabled us to identify three distinct stages in the evolution of melt pools. The strong coupling of the vapour and liquid phases was also evident by this combined analysis, which permitted quantification of the degree of instability according to input energy density. The identified input energy stability threshold of 70 GJ m$^{-3}$ is an upper limit for LPBF processing Ti–6Al–4V. The upper threshold input energy will be different for other materials but could be readily identified with schlieren imaging or any other technique that measures refractive index. A corresponding lower input energy threshold may exist: it is well-known that low energy input can cause hydrodynamic instabilities in the melt pool, resulting in balling or other inconsistencies in solidified tracks[1,39,40], which ultimately lead to lack of fusion defects[10,45]. Although instabilities of that type were observable near the solidification front, the magnification, field of view and sensitivity of our experimental setup were tuned to probe liquid-vapour interactions for this study.

No pores were detected for stages I – II melt pools because of the relative invariance of the LVI, i.e., a stable depression in the melt pool surface. Unstable, stage III keyholes are prone to collapse, potentially resulting in excess porosity[35,46–48]. Increased atmospheric turbulence at $E > 70$ GJ m$^{-3}$ identified that the melt pool had progressed to stage III, corroborated with the depression front wall angle reaching ~70°. This often results in porosity, as can be seen in the supporting materials, for the cases at 68 and 74 GJ m$^{-3}$. In the powder case, where the instability threshold is slightly higher, the onset of porosity occurs between 73 and 81 GJ m$^{-3}$. The demonstrated interconnectivity between liquid and vapour flow dynamics suggests it might be possible to relate porosity formation with features in the turbulent plume; sectioning monitored large area scans and using advanced data processing techniques such as encoder/decoder architectures in convolutional neural networks could allow deeper interpretation of the acquired data, enabling the identification and prevention of defects in real time.

The introduction of powder did not result in significant differences in the observed depression morphology and atmospheric flow. Powder was found to have a mild stabilising effect on the melt pool, with stage III keyholes observed for Ti–6Al–4V just above $E = $ ~70 GJ m$^{-3}$, confirming the validity of the identified threshold as a broad guide for parameter selection. When the plume and the depression were steady, a relatively uniform stream of hot and cold particles trailed the melt pool. Turbulence in the plume resulted in a wider spread in the trajectories of spattered particles, with generally higher velocities due to stronger atmospheric flows, and more frequent particle-jet interactions. These results can aid in identifying trends in existing process monitoring systems, by showcasing the physics underlying cold and hot particle spattering.

Lastly, our experiments also unambiguously illustrate that the laser plume not only ejects powder particles, but also entrains nearby particles, drawing them towards the melt pool. While powder bed denudation[3,43] has been well documented as a phenomenon in LPBF, its effects on the process have only recently started to be explored numerically[25,49,50]. The combined imaging highlights the incorporation of entrained particles into the melt pool, which means that the total molten volume can be significantly increased through the introduction of powder, affecting its stability. Our line scan data show that laser scanning speed and power also affect mass transfer by influencing the intensity and duration of the interaction between the denuding flow and surrounding particles. Additionally, it was clear in the schlieren data that the laser beam often interacts with the ejected vapour and condensate, especially at high laser power density, an effect that is mostly overlooked. Including these effects in numerical models could greatly enhance their predictive capability, allowing a priori calculation of process maps, further facilitating the exploration of new materials and processing methods.

## Methods

**Simultaneous optical and X-ray imaging**. Borosilicate glass windows were used in both sides of the processing chamber, replacing the Kapton windows that are typically used for x-ray imaging: attenuation of the optical beam was too large with Kapton, and its optical thickness is not sufficiently uniform for the requirements of schlieren imaging[19]. The energy of the x-ray flux caused radiation-induced darkening[51] of the glass windows, which attenuated the x-rays and required the exposure time of the camera to be increased, slightly degrading the contrast in the x-ray imaging. However, the darkened spot remained outside of the field of view of the schlieren system and did not affect optical transmissivity. The high-speed camera for x-ray imaging (Photron fastcam SA-Z) was setup at 50 kfps rate and 600 × 640 pixels field-of-view, whilst the camera for schlieren (Photron fastcam mini AX200) recorded at 25 kfps and 384 × 560 pixels resolution. The total magnification for the x-ray setup was four times larger than that of the schlieren system, resulting in approximately 1.2 × 1.3 mm$^2$ and 4.4 × 3 mm$^2$ fields of view, respectively.

**Image processing for display**. All images were processed using MATLAB software, including the image processing toolbox add-on. 16-bit schlieren images from the Fastcam mini camera were initially converted to double precision arrays to facilitate the processing. To reduce the effect of background imperfections due to slight soiling and misalignment of the optics, each frame within the schlieren image sequence was divided by the initial frame, which contained no flow information. Following the division, the brightest pixels had a value ~3, while the darkest were close to 0. Multiplication by a constant (in this case, 85) allowed restoration of the

DC grey levels removed by the division, before direct encoding to 8-bit unsigned integers (0–255 value range). Finally, the contrast of the resulting 8-bit images was stretched by margins fixed across datasets, so that ~2% of pixels were saturated. Background removal was not necessary to adequately display the x-ray images. Thus, images were imported to MATLAB directly as 16-bit, contrast stretched to saturate ~2% of pixels (fixed margins across datasets) and converted to 8-bit.

Videos of each processed image sequence were produced in MATLAB. Composite videos were produced by importing the individual videos into OpenShot video editor software. The schlieren videos were upscaled by a factor of 2, while the X-ray videos were downscaled by a factor of 0.5. In the overlaid region, an x-ray/schlieren intensity ratio of 0.6 was found to yield the best result.

**Image differencing**. The main advantages of this approach are its simplicity, and that both schlieren and x-ray datasets were evaluated using a common metric. Images of the intensity difference between one frame and the previous were produced, in which each pixel takes the value $p_i$ defined as $p_{i,k} = \left( X_{i,k} - X_{i-1,k} \right)^2$, where $X_{i,k}$ is the intensity value of the $k$th pixel in the $i$th frame of the sequence. For spot illumination, the total difference $\sum p$ in each timestep, simply taken as the sum of all pixels of every frame and normalised by division with the maximum value in the dataset, was used in Fig. 3. For line scan illumination, to reduce the information for the entire image sequence to a single point average $\overline{\sum p}$, the total difference of every frame in which the laser was on was summed up and divided by the number of such frames, as plotted in Fig. 5.

**Keyhole boundary detection**. Initially, 16-bit x-ray images were cropped to only include the upper ~200 µm of substrate. Background removal by division, direct conversion to 8-bit unsigned integers and contrast stretching followed, as outlined above. To reduce noise, images were filtered by convolution with a 2D Gaussian kernel with a standard deviation of 3, in addition to median filtering with a $7 \times 7$ pixel kernel. The filtered images were then binarised, based on a global threshold. Interestingly, trial and error with 2–3 different values for the binarisation threshold (0.48–0.5 in this case) produced better results than more sophisticated adaptive thresholding techniques, such as Otsu's method. The edges of the binary image were then detected using the Sobel edge detection method. For some frames, small imperfections in the substrate or camera noise would remain above the binarisation threshold and therefore register as distinct edges. This was corrected by counting the number of pixels within each detected area and considering only the one with highest total. The detected edge was then overlaid onto the original image for display. The standard deviation of the detected area was calculated as

$\sigma = \sqrt{\left( A - \mu \right)^2 / (N - 1)}$, where $A$ is the keyhole area in each frame and $\mu$ is the mean area over $N$ analysed frames.

## Data availability

Video files corresponding to the composite image figures of this article are included in the supplementary information files. The datasets generated and/or analysed during the current study are available in the PURE repository, with the identifier https://doi.org/10.17861/79a2a7ae-a3f5-47ad-bb2a-94dddcb32084.

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

## Acknowledgements
The authors are grateful to Tim Nicholls of Photron Ltd. for use of the Photron Fastcam Mini AX200 camera. I.B. and A.J.M. acknowledge support by the Engineering and Physical Sciences Research Council (Grant number EP/P027415/1) and Renishaw plc. A.D.R. acknowledges support from the NASA ULI program under grant number 80NSSC19M0123. This research used resources of the Advanced Photon Source, a U.S. Department of Energy (DOE) Office of Science User Facility operated for the DOE Office of Science by Argonne National Laboratory under Contract No. DE-AC02-06CH11357.

## Author contributions
I.B. wrote the manuscript, with contributions and editing by T.S., A.D.R. and A.J.M. I.B., N.P., C.Z. and T.S. conducted the experiments at the Advanced Photon Source. I.B. performed the image processing. I.B. and A.J.M. carried out the formal analysis of the results. The work was supervised by T.S., A.D.R. and A.J.M.

## Competing interests
The authors declare no competing interests
