## [Peer Review File · Nature Communications]

Title: The interplay between vapour, liquid, and solid phases in laser powder bed fusionReviewers' comments:

Reviewer #1 (Remarks to the Author):

Review of Bitharas et al., "The interplay between vapour, liquid, and solid phases in laser powder bed fusion", MS# NCOMMS-21-26277

The authors present a combined study of laser powder bed fusion process dynamics using high speed optical schlieren imaging and high-speed x-ray radiographic imaging. This combined imaging approach allows them to reveal the interplay between melt pool motion and plume instabilities in a way that has not yet been presented in the community. The laminar to turbulent transition is very nicely documented and is correlated succinctly with vapor depression interface motion. However, despite the novelty of combining these techniques, the insights gained do not seem to represent any breakthrough in understanding or exposing new phenomena and appear to mainly confirm the analysis presented in the great number of in situ x-ray radiographic measurements to date. Some of the arguments put forth interpreting the data appear hard to understand or (in one case, see below) perhaps unfounded. Still, the work presented offers additional data with which to validate full CFD + contact mechanics simulations which are still needed (as the authors point out) to more completely describe the mesoscale physics of LPBF.

In addition to the comments above, please consider the following:

-line 68: should specify 'co-axial' as referring either to x-ray or optical path

-fig 2a: the 'toroidal vortex' lacks clear definition in the figure, and description in the text could be stronger to help the reader understand what in the figure is distinctive.

-line 128: how can an increase in temperature and concentration be claimed if only gradients are detected?

-fig 3 / methods: since the figure presents instabilities, it is important to affirm the stability of the laser power, particularly at turn on. Can the authors comment on the output stability of the laser?

Lines 182-184: the authors are discussing scaling behavior for laser power and scan velocity; the sentence here however is worded awkwardly and should be rewritten.

Line 205-206: "The melt pool evolves to stage II..." it's unclear what is being referred to in figure 4, please explain in clearer fashion.

Line 247: it's unclear what "Separate gradient" is supposed to mean.

-fig 6: it seems that (a) and (b) are missing legends for symbols, though it is assumed they are the same as in (c).

Lines 264-265: there seem to be an inconsistency in the discussion of transitions in terms of 'stages' and keyhole-conduction, despite noting earlier that the authors wished to 'avoid' using the latter terms.

Line 266: the idea that the energy is 'seemingly dissipated' seems unfounded. It is well established that the transition from no or small vapor depression to a deep keyhole depression leads to an increase in absorptivity and the effect on melt pool depth (and vapor depression) have been documented. Can the authors estimate what the change in absorptivity is expected to be over the range discussed here?

Line 298-301: the blocking of light is straight-forward to understand but the argument about loosely

packed powder is unclear.

Reviewer #2 (Remarks to the Author):

The authors report on the simultaneous use of in-situ synchrotron x-ray and schlieren imaging in laser powder-bed fusion to detect fluid dynamics of the vapor jet formed during the process and the melt pool dynamics.

The study is novel and outstanding. A lot of work is done in this study, and I applaud it. However, there is a fundamental/philosophical issue with the experimental approach/setup that does not allow me to recommend this paper for publication in the prestigious journal of Nature Communication.

Here are my concerns:

- 1) Gas flow is one of the main factors in laser powder-bed fusion (LBBF). In LPBF, a laminar gas stream carries on top of the powder bed to not only remove the plume and spatters from the process zone, but also minimize the smoking phenomenon and stabilize the melt pool dynamics. In this study, this gas flow has not been incorporated into the testing setup, and a stationary Ar atmosphere is considered. Thus the process of interest in this study does not mockup LBBF. Major skewness in the plume/vapor flow will be created due to the incorporated gas flow. Any results identified in this study may not be extendable to the actual LPBF process.
- 2) All phenomena explained in this paper, e.g., formation of toroidal vortex, recoil pressure, vapor jet dissipation, plume cap, etc. are highly affected by the gas flow, passing on top of the powder-bed, in a real LPBF process.
- 3) There is no analysis based on Reynolds number to quantify laminar and turbulent flows scientifically.
- 4) The compaction density of pre-placed powder is not given. This factor affects the melt pool dynamics and plume formation significantly.

Reviewer #3 (Remarks to the Author):

This research employs simultaneous in-situ synchrotron x-ray and schlieren imaging, in order to image the interconnected fluid dynamics of the vapor jet formed by the laser and the melt pool depression. Process instabilities are analyzed across several parameter sets by measuring keyhole and plume morphologies and fluctuations thereof. The authors have identified a previously unreported threshold of the energy input required for stable line scans with Ti-6Al-4V alloy.

This research appears to be the first of its kind using combined x-ray and schlieren imaging to

simultaneously visualize dynamics within the metallic keyhole and ejected gas/vapor plume. The data are of high-quality both in spatial and temporal resolution and are produced under processing parameters (laser power, spot size, scan speed) of interest for Ti-6Al-4V LPBF. The image processing approaches employed are generally straightforward and robust—subsequent image differencing for x-ray images as well as schlieren images as the primary measure of keyhole and plume stability. The analytical approaches employed for input energy density, power density, and front wall angle are well grounded in recent and relevant literature. The manuscript is well written and supported with adequate motivation for the research effort and reporting thereof.

No further changes to the experimental or data processing approaches are recommended, but the following changes may improve the validity and clarity of the descriptions of the reported process physics:

1. The authors state in line 51 that one of the primary outcomes of this work is that “We reveal that the onset of capillary instability within the keyhole causes a transition from laminar to turbulent flow in the laser plume.” In traditional fluid jet theory, a laminar fluid jet may transition to turbulence at some distance from the nozzle due to growing instabilities that result in eddy formation and enhanced mixing with the surrounding fluid. Alternatively, increased Reynolds number may result in a transition to turbulence at the nozzle outlet. In this paper, the authors seem to claim that instability in the keyhole causes rapid changes in the jet propagation direction that results in a transition to turbulence. A laminar jet with a fluctuation propagation direction may not meet the traditional definition of turbulence, and so the authors should provide further explanation as to why the flow should be considered turbulent. If there is insufficient evidence that the Reynolds number causes the transition from the laminar to “turbulent” cases, the authors should consider relabeling the “turbulent” cases to, perhaps, “chaotic” or “fluctuating.”

2. The authors statement in line 51 that “We reveal that the onset of capillary instability within the keyhole causes a transition from laminar to turbulent flow in the laser plume” indicates a clear cause-effect relationship, with the capillary instabilities causing the transition to chaotic plume behavior. Yet on line 330, the authors state that “Where the laser is incident, the recoil pressure pushing the surface is in competition with the local Marangoni force and capillary pressure.” In cases where a significant keyhole is established, the vapor forces are much stronger than in lower energy density cases where a globular melt pool can be formed. And so, it seems likely that the interplay between vapor recoil and capillary instabilities compound to result in the transition to a chaotic plume. The authors should provide further evidence/explanation that capillary instabilities are the cause of the change in the plume or avoid claiming such a clear cause-effect relationship.

3. On line 84 the authors state that there is “a steady jet of Al, V and Ti vapours.” It is quite possible that the hottest point of the melt pool does not exceed 3400 °C, so V may not necessarily be vaporized. The elements will also certainly be vaporized at different rates. So, if the constituents of the vapor jet are not directly known, the authors should consider rewording this statement or elaborating on it.

4. The authors do not seem to make any mention of directional crossflow of the inert gas, which exists in essentially all commercial LPBF machines. Such horizontal crossflow provisions are implemented to carry away the plume of process byproducts that is known to interfere with laser delivery. Lack of such flow is known to attenuate and scatter the incoming laser beam, which further complicates the interplay between the laser, vapor, capillarity, and phase change. The authors should justify why the results of these experiments apply to LPBF machines, or clearly state that these results are only valid for applications with an absence of crossflow.

5. It is recommended for the authors to discuss the powder size distribution used in these experiments for thoroughness.

6. The authors should state the process laser wavelength.

Terminology and consistency items:

1. All instances of “interphase” may need to be changed to “interface”
2. On line 39, use of “printing” should be changed to “additive manufacturing” for consistency/clarity
3. Use of “laser plume” seems to be an uncommon term; “process plume” may be more appropriate
4. On line 203 the authors state that the “keyhole is deeper and wider (along the scanning direction).” The authors may consider rewording to the “keyhole is deeper and longer.” More common terminology is that length is along the scan direction and width is perpendicular to the scan direction.
5. On line 290 the authors state that the process “ejects metal and particles alike.” They should consider a clarification such as “ejects spatter and particles alike”
6. On line 316: “The molten sphere” is an inaccurate description because the molten material is certainly not spherical. Perhaps “The molten mass” would be more appropriate.
7. On line 383: “the behaviour of the surface depression is mirrored in the motion of the vapour jet.” This is a very non-technical use of the word “mirrored”—especially considering that literal mirrors are described in the schlieren setup. Perhaps “shown” or “indicated” would be more appropriate.

Response to reviewers and list of changes

The authors would like to thank the reviewers for the time and effort they put into thoroughly reviewing our work. In addition to including a new section “Influence of Ar cross-flow” to the manuscript, we have individually addressed each reviewer’s comment (highlighted in yellow below) as follows:

Reviewer 1

-line 68: should specify ‘co-axial’ as referring either to x-ray or optical path.

Added “(to the x-ray beam)” in the text.

-fig 2a: the ‘toroidal vortex’ lacks clear definition in the figure, and description in the text could be stronger to help the reader understand what in the figure is distinctive.

Restructured text and added a sentence to elaborate further.

-line 128: how can an increase in temperature and concentration be claimed if only gradients are detected?

In that sentence, we explain that the refractive index of the background remains constant, and therefore the stronger gradients must translate to higher or lower refractive index in the plume and thus higher temperature and fume/vapour content.

-fig 3 / methods: since the figure presents instabilities, it is important to affirm the stability of the laser power, particularly at turn on. Can the authors comment on the output stability of the laser?

The maximum power fluctuation of the laser was ± 0.7 W at this power setting, which would not be sufficient to cause an appreciable effect. The rise time was a maximum of 20 μ s, which would be the equivalent of 1 frame in the x-ray footage and would therefore not influence the imaged progression towards instability. We have included this information and measurements of the laser’s output stability in the supplementary material.

Lines 182-184: the authors are discussing scaling behaviour for laser power and scan velocity; the sentence here however is worded awkwardly and should be rewritten.

We have reworded the sentence and removed the awkward part.

Line 205-206: “The melt pool evolves to stage II...” it’s unclear what is being referred to in figure 4, please explain in clearer fashion.

We have reworded the sentence for clarity.

Line 247: it’s unclear what “Separate gradient” is supposed to mean.

We change “Separate gradient” to “different line” to clarify.

-fig 6: it seems that (a) and (b) are missing legends for symbols, though it is assumed they are the same as in (c).

Indeed, this was done to avoid cluttering the graphs with three identical legends. We have clarified this in the text describing the figure.

Lines 264-265: there seem to be an inconsistency in the discussion of transitions in terms of ‘stages’ and keyhole-conduction, despite noting earlier that the authors wished to ‘avoid’ using the latter terms.

The text now writes “deeper keyhole” to avoid confusion with the keyhole regime.

Line 266: the idea that the energy is ‘seemingly dissipated’ seems unfounded. It is well established that the transition from no or small vapor depression to a deep keyhole depression leads to an increase in absorptivity and the effect on melt pool depth (and vapor depression) have been documented. Can the authors estimate what the change in absorptivity is expected to be over the range discussed here?

We have estimated the change in absorptivity and absorbed energy for the input energy density range discussed here, according to reference [32]. As the original measurements in that work were not presented in terms of energy density, we include two new plots in the supplementary materials based on their data. Also, we have rephrased our point to avoid confusion.

Line 298-301: the blocking of light is straight-forward to understand but the argument about loosely packed powder is unclear.

We have reworded the sentence to clarify the argument further.

Reviewer 2

Points 1) and 2) are addressed in the new section “Influence of Ar cross-flow”.

3) There is no analysis based on Reynolds number to quantify laminar and turbulent flows scientifically.

We have added a paragraph discussing the Reynolds number and turbulence, including an estimate. As our experiments, analyses and scientific literature to date do not provide enough evidence to characterise the flow in this manner, we have indeed relabelled all “turbulent” cases to “chaotic” or “unstable” and all “laminar” cases to “stable” throughout the manuscript.

4) The compaction density of pre-placed powder is not given. This factor affects the melt pool dynamics and plume formation significantly.

We now state that the powder was spread manually with no compaction.

Reviewer 3

The authors state in line 51 that one of the primary outcomes of this work is that “We reveal that the onset of capillary instability within the keyhole causes a transition from laminar to turbulent flow in the laser plume.” In traditional fluid jet theory, a laminar fluid jet may transition to turbulence at some distance from the nozzle due to growing instabilities that result in eddy formation and enhanced mixing with the surrounding fluid. Alternatively, increased Reynolds number may result in a transition to turbulence at then nozzle outlet. In this paper, the authors seem to claim that instability in the keyhole causes rapid changes in the jet propagation direction that results in a transition to turbulence. A laminar jet with a fluctuation propagation direction may not meet the traditional definition of turbulence, and so the authors should provide further explanation as to why the flow should be

considered turbulent. If there is insufficient evidence that the Reynolds number causes the transition from the laminar to “turbulent” cases, the authors should consider relabeling the “turbulent” cases to, perhaps, “chaotic” or “fluctuating.”

Same as point 3) raised by reviewer 2, which we address in the additional paragraph described there.

2. The authors statement in line 51 that “We reveal that the onset of capillary instability within the keyhole causes a transition from laminar to turbulent flow in the laser plume” indicates a clear cause-effect relationship, with the capillary instabilities causing the transition to chaotic plume behavior. Yet on line 330, the authors state that “Where the laser is incident, the recoil pressure pushing the surface is in competition with the local Marangoni force and capillary pressure.” In cases where a significant keyhole is established, the vapor forces are much stronger than in lower energy density cases where a globular melt pool can be formed. And so, it seems likely that the interplay between vapor recoil and capillary instabilities compound to result in the transition to a chaotic plume. The authors should provide further evidence/explanation that capillary instabilities are the cause of the change in the plume or avoid claiming such a clear cause-effect relationship.

We have changed the text on line 51 and elsewhere in the manuscript to acknowledge the contribution of recoil pressure in addition to capillary instabilities.

3. On line 84 the authors state that there is “a steady jet of Al, V and Ti vapours.” It is quite possible that the hottest point of the melt pool does not exceed 3400 °C, so V may not necessarily be vaporized. The elements will also certainly be vaporized at different rates. So, if the constituents of the vapor jet are not directly known, the authors should consider rewording this statement or elaborating on it.

We agree and have reworded the sentence for clarity and added a reference from a recent paper on Ti64 evaporation to guide the reader.

4. is addressed in the new section “Influence of Ar cross-flow”.

5. It is recommended for the authors to discuss the powder size distribution used in these experiments for thoroughness.

The size distribution is now included in the text.

6. The authors should state the process laser wavelength.

Laser wavelength now included in text.

Terminology and consistency items:

1. All instances of “interphase” may need to be changed to “interface”
2. On line 39, use of “printing” should be changed to “additive manufacturing” for consistency/clarity
3. Use of “laser plume” seems to be an uncommon term; “process plume” may be more appropriate
4. On line 203 the authors state that the “keyhole is deeper and wider (along the scanning direction).” The authors

may consider rewording to the “keyhole is deeper and longer.” More common terminology is that length is along the scan direction and width is perpendicular to the scan direction.

5. On line 290 the authors state that the process “ejects metal and particles alike.” They should consider a clarification such as “ejects spatter and particles alike”

6. On line 316: “The molten sphere” is an inaccurate description because the molten material is certainly not spherical. Perhaps “The molten mass” would be more appropriate.

7. On line 383: “the behaviour of the surface depression is mirrored in the motion of the vapour jet.” This is a very non-technical use of the word “mirrored”—especially considering that literal mirrors are described in the schlieren setup. Perhaps “shown” or “indicated” would be more appropriate.

We agree with all proposed terminology changes apart from the use of “laser plume”, which we believe is an unambiguous term with sufficient precedent in the literature. We have edited the manuscript accordingly.

REVIEWERS' COMMENTS

Reviewer #1 (Remarks to the Author):

The authors have addressed my concerns.

Reviewer #2 (Remarks to the Author):

The authors have done a great job in addressing my concerns. In particular, they have added a new section on the role of laminar gas flow on the plume mechanism in LPBF.

I am happy with this revision and it can proceed with publication.

Reviewer #3 (Remarks to the Author):

The authors have adequately addressed my concerns. The manuscript is recommended for publication in Nature Communications.